# Differentiable Convex Optimization Layers

**Akshay Agrawal**
Stanford University
akshayka@cs.stanford.edu

**Brandon Amos**
Facebook AI
bda@fb.com

**Shane Barratt**
Stanford University
sbarratt@stanford.edu

**Stephen Boyd**
Stanford University
boyd@stanford.edu

**Steven Diamond**
Stanford University
diamond@cs.stanford.edu

**J. Zico Kolter**[*]
Carnegie Mellon University
Bosch Center for AI
zkolter@cs.cmu.edu

## Abstract

Recent work has shown how to embed *differentiable optimization problems* (that is, problems whose solutions can be backpropagated through) as layers within deep learning architectures. This method provides a useful inductive bias for certain problems, but existing software for differentiable optimization layers is rigid and difficult to apply to new settings. In this paper, we propose an approach to differentiating through disciplined convex programs, a subclass of convex optimization problems used by domain-specific languages (DSLs) for convex optimization. We introduce *disciplined parametrized programming*, a subset of disciplined convex programming, and we show that every disciplined parametrized program can be represented as the composition of an affine map from parameters to problem data, a solver, and an affine map from the solver's solution to a solution of the original problem (a new form we refer to as *affine-solver-affine* form). We then demonstrate how to efficiently differentiate through each of these components, allowing for end-to-end analytical differentiation through the entire convex program. We implement our methodology in version 1.1 of CVXPY, a popular Python-embedded DSL for convex optimization, and additionally implement differentiable layers for disciplined convex programs in PyTorch and TensorFlow 2.0. Our implementation significantly lowers the barrier to using convex optimization problems in differentiable programs. We present applications in linear machine learning models and in stochastic control, and we show that our layer is competitive (in execution time) compared to specialized differentiable solvers from past work.

## 1   Introduction

Recent work has shown how to differentiate through specific subclasses of convex optimization problems, which can be viewed as functions mapping problem data to solutions [6, 31, 10, 1, 4]. These layers have found several applications [40, 6, 35, 27, 5, 53, 75, 52, 12, 11], but many applications remain relatively unexplored (see, *e.g.*, [4, §8]).

While convex optimization layers can provide useful inductive bias in end-to-end models, their adoption has been slowed by how difficult they are to use. Existing layers (*e.g.*, [6, 1]) require users to transform their problems into rigid canonical forms by hand. This process is tedious, error-prone, and time-consuming, and often requires familiarity with convex analysis. Domain-specific languages (DSLs) for convex optimization abstract away the process of converting problems to canonical forms, letting users specify problems in a natural syntax; programs are then lowered to canonical forms and

---

[*]Authors listed in alphabetical order.

supplied to numerical solvers behind-the-scenes [3]. DSLs enable rapid prototyping and make convex optimization accessible to scientists and engineers who are not necessarily experts in optimization.

The point of this paper is to do what DSLs have done for convex optimization, but for differentiable convex optimization layers. In this work, we show how to efficiently differentiate through disciplined convex programs [45]. This is a large class of convex optimization problems that can be parsed and solved by most DSLs for convex optimization, including CVX [44], CVXPY [29, 3], Convex.jl [72], and CVXR [39]. Concretely, we introduce *disciplined parametrized programming* (DPP), a grammar for producing parametrized disciplined convex programs. Given a program produced by DPP, we show how to obtain an affine map from parameters to problem data, and an affine map from a solution of the canonicalized problem to a solution of the original problem. We refer to this representation of a problem — *i.e.*, the composition of an affine map from parameters to problem data, a solver, and an affine map to retrieve a solution — as *affine-solver-affine* (ASA) form.

Our contributions are three-fold:

**1.** We introduce DPP, a new grammar for parametrized convex optimization problems, and ASA form, which ensures that the mapping from problem parameters to problem data is affine. DPP and ASA-form make it possible to differentiate through DSLs for convex optimization, *without explicitly backpropagating through the operations of the canonicalizer*. We present DPP and ASA form in §4.

**2.** We implement the DPP grammar and a reduction from parametrized programs to ASA form in CVXPY 1.1. We also implement differentiable convex optimization layers in PyTorch [66] and TensorFlow 2.0 [2]. Our software substantially lowers the barrier to using convex optimization layers in differentiable programs and neural networks (§5).

**3.** We present applications to sensitivity analysis for linear machine learning models, and to learning control-Lyapunov policies for stochastic control (§6). We also show that for quadratic programs (QPs), our layer's runtime is competitive with OptNet's specialized solver `qpth` [6] (§7).

## 2 Related work

**DSLs for convex optimization.** DSLs for convex optimization allow users to specify convex optimization problems in a natural way that follows the math. At the foundation of these languages is a ruleset from convex analysis known as disciplined convex programming (DCP) [45]. A mathematical program written using DCP is called a disciplined convex program, and all such programs are convex. Disciplined convex programs can be *canonicalized* to cone programs by expanding each nonlinear function into its graph implementation [43]. DPP can be seen as a subset of DCP that mildly restricts the way parameters (symbolic constants) can be used; a similar grammar is described in [26]. The techniques used in this paper to canonicalize parametrized programs are similar to the methods used by code generators for optimization problems, such as CVXGEN [60], which targets QPs, and QCML, which targets second-order cone programs (SOCPs) [26, 25].

**Differentiation of optimization problems.** Convex optimization problems do not in general admit closed-form solutions. It is nonetheless possible to differentiate through convex optimization problems by implicitly differentiating their optimality conditions (when certain regularity conditions are satisfied) [36, 68, 6]. Recently, methods were developed to differentiate through convex cone programs in [24, 1] and [4, §7.3]. Because every convex program can be cast as a cone program, these methods are general. The software released alongside [1], however, requires users to express their problems in conic form. Expressing a convex optimization problem in conic form requires a working knowledge of convex analysis. Our work abstracts away conic form, letting the user differentiate through high-level descriptions of convex optimization problems; we canonicalize these descriptions to cone programs on the user's behalf. This makes it possible to rapidly experiment with new families of differentiable programs, induced by different kinds of convex optimization problems.

Because we differentiate through a cone program by implicitly differentiating its solution map, our method can be paired with any algorithm for solving convex cone programs. In contrast, methods that differentiate through every step of an optimization procedure must be customized for each algorithm (*e.g.*, [33, 30, 56]). Moreover, such methods only approximate the derivative, whereas we compute it analytically (when it exists).

# 3 Background

**Convex optimization problems.** A parametrized convex optimization problem can be represented as

$$
\begin{array}{ll}
\text{minimize} & f_0(x; \theta) \\
\text{subject to} & f_i(x; \theta) \leq 0, \quad i = 1, \ldots, m_1, \\
& g_i(x; \theta) = 0, \quad i = 1, \ldots, m_2,
\end{array}
\tag{1}
$$

where $x \in \mathbf{R}^n$ is the optimization variable and $\theta \in \mathbf{R}^p$ is the parameter vector [22, §4.2]. The functions $f_i : \mathbf{R}^n \to \mathbf{R}$ are convex, and the functions $g_i : \mathbf{R}^n \to \mathbf{R}$ are affine. A *solution* to (1) is any vector $x^\star \in \mathbf{R}^n$ that minimizes the objective function, among all choices that satisfy the constraints. The problem (1) can be viewed as a (possibly multi-valued) function that maps a parameter to solutions. In this paper, we consider the case when this solution map is single-valued, and we denote it by $\mathcal{S} : \mathbf{R}^p \to \mathbf{R}^n$. The function $S$ maps a parameter $\theta$ to a solution $x^\star$. From the perspective of end-to-end learning, $\theta$ (or parameters it depends on) is learned in order to minimize some scalar function of $x^\star$. In this paper, we show how to obtain the derivative of $\mathcal{S}$ with respect to $\theta$, when (1) is a DPP-compliant program (and when the derivative exists).

We focus on convex optimization because it is a powerful modeling tool, with applications in control [20, 16, 71], finance [57, 19], energy management [63], supply chain [17, 15], physics [51, 8], computational geometry [73], aeronautics [48], and circuit design [47, 21], among other fields.

**Disciplined convex programming.** DCP is a grammar for constructing convex optimization problems [45, 43]. It consists of functions, or *atoms*, and a single rule for composing them. An atom is a function with known curvature (affine, convex, or concave) and per-argument monotonicities. The composition rule is based on the following theorem from convex analysis. Suppose $h : \mathbf{R}^k \to \mathbf{R}$ is convex, nondecreasing in arguments indexed by a set $I_1 \subseteq \{1, 2, \ldots, k\}$, and nonincreasing in arguments indexed by $I_2$. Suppose also that $g_i : \mathbf{R}^n \to \mathbf{R}$ are convex for $i \in I_1$, concave for $i \in I_2$, and affine for $i \in (I_1 \cap I_2)^c$. Then the composition $f(x) = h(g_1(x), g_2(x), \ldots, g_k(x))$ is convex. DCP allows atoms to be composed so long as the composition satisfies this composition theorem. Every disciplined convex program is a convex optimization problem, but the converse is not true. This is not a limitation in practice, because atom libraries are extensible (*i.e.*, the class corresponding to DCP is parametrized by which atoms are implemented). In this paper, we consider problems of the form (1) in which the functions $f_i$ and $g_i$ are constructed using DPP, a version of DCP that performs parameter-dependent curvature analysis (see §4.1).

**Cone programs.** A (convex) cone program is an optimization problem of the form

$$
\begin{array}{ll}
\text{minimize} & c^T x \\
\text{subject to} & b - Ax \in \mathcal{K},
\end{array}
\tag{2}
$$

where $x \in \mathbf{R}^n$ is the variable (there are several other equivalent forms for cone programs). The set $\mathcal{K} \subseteq \mathbf{R}^m$ is a nonempty, closed, convex cone, and the *problem data* are $A \in \mathbf{R}^{m \times n}$, $b \in \mathbf{R}^m$, and $c \in \mathbf{R}^n$. In this paper we assume that (2) has a unique solution.

Our method for differentiating through disciplined convex programs requires calling a solver (an algorithm for solving an optimization problem) in the forward pass. We focus on the special case in which the solver is a *conic solver*. A conic solver targets convex cone programs, implementing a function $s : \mathbf{R}^{m \times n} \times \mathbf{R}^m \times \mathbf{R}^n \to \mathbf{R}^n$ mapping the problem data $(A, b, c)$ to a solution $x^\star$.

DCP-based DSLs for convex optimization can canonicalize disciplined convex programs to equivalent cone programs, producing the problem data $A, b, c$, and $\mathcal{K}$ [3]; $(A, b, c)$ depend on the parameter $\theta$ and the canonicalization procedure. These data are supplied to a conic solver to obtain a solution; there are many high-quality implementations of conic solvers (*e.g.*, [64, 9, 32]).

# 4 Differentiating through disciplined convex programs

We consider a disciplined convex program with variable $x \in \mathbf{R}^n$, parametrized by $\theta \in \mathbf{R}^p$; its solution map can be viewed as a function $\mathcal{S} : \mathbf{R}^p \to \mathbf{R}^n$ that maps parameters to the solution (see §3). In this section we describe the form of $\mathcal{S}$ and how to evaluate $\mathsf{D}^T \mathcal{S}$, allowing us to backpropagate through parametrized disciplined convex programs. (We use the notation $\mathsf{D}f(x)$ to denote the derivative of

a function $f$ evaluated at $x$, and $\mathsf{D}^T f(x)$ to denote the adjoint of the derivative at $x$.) We consider the special case of canonicalizing a disciplined convex program to a cone program. With little extra effort, our method can be extended to other targets.

We express $\mathcal{S}$ as the composition $R \circ s \circ C$; the canonicalizer $C$ maps parameters to cone problem data $(A, b, c)$, the cone solver $s$ solves the cone problem, furnishing a solution $\tilde{x}^\star$, and the retriever $R$ maps $\tilde{x}^\star$ to a solution $x^\star$ of the original problem. A problem is in ASA form if $C$ and $R$ are affine.

By the chain rule, the adjoint of the derivative of a disciplined convex program is

$$\mathsf{D}^T \mathcal{S}(\theta) = \mathsf{D}^T C(\theta) \mathsf{D}^T s(A, b, c) \mathsf{D}^T R(\tilde{x}^\star).$$

The remainder of this section proceeds as follows. In §4.1, we present DPP, a ruleset for constructing disciplined convex programs reducible to ASA form. In §4.2, we describe the canonicalization procedure and show how to represent $C$ as a sparse matrix. In §4.3, we review how to differentiate through cone programs, and in §4.4, we describe the form of $R$.

## 4.1 Disciplined parametrized programming

DPP is a grammar for producing parametrized disciplined convex programs from a set of functions, or atoms, with known curvature (constant, affine, convex, or concave) and per-argument monotonicities. A program produced using DPP is called a disciplined parametrized program. Like DCP, DPP is based on the well-known composition theorem for convex functions, and it guarantees that every function appearing in a disciplined parametrized program is affine, convex, or concave. Unlike DCP, DPP also guarantees that the produced program can be reduced to ASA form.

A disciplined parametrized program is an optimization problem of the form

$$\begin{array}{ll} \text{minimize} & f_0(x, \theta) \\ \text{subject to} & f_i(x, \theta) \leq \tilde{f}_i(x, \theta), \quad i = 1, \ldots, m_1, \\ & g_i(x, \theta) = \tilde{g}_i(x, \theta), \quad i = 1, \ldots, m_2, \end{array} \tag{3}$$

where $x \in \mathbf{R}^n$ is a variable, $\theta \in \mathbf{R}^p$ is a parameter, the $f_i$ are convex, $\tilde{f}_i$ are concave, $g_i$ and $\tilde{g}_i$ are affine, and the expressions are constructed using DPP. An expression can be thought of as a tree, where the nodes are atoms and the leaves are variables, constants, or parameters. A parameter is a symbolic constant with known properties such as sign but unknown numeric value. An expression is said to be parameter-affine if it does not have variables among its leaves and is affine in its parameters; an expression is parameter-free if it is not parametrized, and variable-free if it does not have variables.

Every DPP program is also DCP, but the converse is not true. DPP generates programs reducible to ASA form by introducing two restrictions on expressions involving parameters:

1. In DCP, we classify the curvature of each subexpression appearing in the problem description as convex, concave, affine, or constant. All parameters are classified as constant. In DPP, parameters are classified as affine, just like variables.

2. In DCP, the product atom $\phi_{\text{prod}}(x, y) = xy$ is affine if $x$ or $y$ is a constant (*i.e.*, variable-free). Under DPP, the product is affine when at least one of the following is true:

   - $x$ or $y$ is constant (*i.e.*, both parameter-free and variable-free);
   - one of the expressions is parameter-affine and the other is parameter-free.

The DPP specification can (and may in the future) be extended to handle several other combinations of expressions and parameters.

**Example.** Consider the program

$$\begin{array}{ll} \text{minimize} & \|Fx - g\|_2 + \lambda \|x\|_2 \\ \text{subject to} & x \geq 0, \end{array} \tag{4}$$

with variable $x \in \mathbf{R}^n$ and parameters $F \in \mathbf{R}^{m \times n}$, $g \in \mathbf{R}^m$, and $\lambda > 0$. If $\|\cdot\|_2$, the product, negation, and the sum are atoms, then this problem is DPP-compliant:

- $\phi_{\text{prod}}(F, x) = Fx$ is affine because the atom is affine ($F$ is parameter-affine and $x$ is parameter-free) and $F$ and $x$ are affine;

- $Fx - g$ is affine because $Fx$ and $-g$ are affine and the sum of affine expressions is affine;

- $\|Fx - g\|_2$ is convex because $\|\cdot\|_2$ is convex and convex composed with affine is convex;

- $\phi_{\mathrm{prod}}(\lambda, \|x\|_2)$ is convex because the product is affine ($\lambda$ is parameter-affine, $\|x\|_2$ is parameter-free), it is increasing in $\|x\|_2$ (because $\lambda$ is nonnegative), and $\|x\|_2$ is convex;

- the objective is convex because the sum of convex expressions is convex.

**Non-DPP transformations of parameters.** It is often possible to re-express non-DPP expressions in DPP-compliant ways. Consider the following examples, in which the $p_i$ are parameters:

- The expression $\phi_{\mathrm{prod}}(p_1, p_2)$ is not DPP because both of its arguments are parametrized. It can be rewritten in a DPP-compliant way by introducing a variable $s$, replacing $p_1 p_2$ with the expression $p_1 s$, and adding the constraint $s = p_2$.

- Let $e$ be an expression. The quotient $e/p_1$ is not DPP, but it can be rewritten as $e p_2$, where $p_2$ is a new parameter representing $1/p_1$.

- The expression $\log |p_1|$ is not DPP because $\log$ is concave and increasing but $|\cdot|$ is convex. It can be rewritten as $\log p_2$ where $p_2$ is a new parameter representing $|p_1|$.

- If $P_1 \in \mathbf{R}^{n \times n}$ is a parameter representing a (symmetric) positive semidefinite matrix and $x \in \mathbf{R}^n$ is a variable, the expression $\phi_{\mathrm{quadform}}(x, P_1) = x^T P_1 x$ is not DPP. It can be rewritten as $\|P_2 x\|_2^2$, where $P_2$ is a new parameter representing $P_1^{1/2}$.

## 4.2   Canonicalization

The canonicalization of a disciplined parametrized program to ASA form is similar to the canonicalization of a disciplined convex program to a cone program. All nonlinear atoms are expanded into their graph implementations [43], generating affine expressions of variables. The resulting expressions are also affine in the problem parameters due to the DPP rules. Because these expressions represent the problem data for the cone program, the function $C$ from parameters to problem data is affine.

As an example, the DPP program (4) can be canonicalized to the cone program

$$
\begin{aligned}
\text{minimize} \quad & t_1 + \lambda t_2 \\
\text{subject to} \quad & (t_1, Fx - g) \in \mathcal{Q}_{m+1}, \\
& (t_2, x) \in \mathcal{Q}_{n+1}, \\
& x \in \mathbf{R}_+^n,
\end{aligned}
\tag{5}
$$

where $(t_1, t_2, x)$ is the variable, $\mathcal{Q}_n$ is the $n$-dimensional second-order cone, and $\mathbf{R}_+^n$ is the nonnegative orthant. When rewritten in the standard form (2), this problem has data

$$
A = \begin{bmatrix} -1 & & \\ & -F & \\ \hline & -1 & \\ & & -I \\ \hline & & -I \end{bmatrix}, \quad b = \begin{bmatrix} 0 \\ -g \\ 0 \\ 0 \\ 0 \end{bmatrix}, \quad c = \begin{bmatrix} 1 \\ \lambda \\ 0 \end{bmatrix}, \quad \mathcal{K} = \mathcal{Q}_{m+1} \times \mathcal{Q}_{n+1} \times \mathbf{R}_+^n,
$$

with blank spaces representing zeros and the horizontal line denoting the cone boundary. In this case, the parameters $F$, $g$ and $\lambda$ are just negated and copied into the problem data.

**The canonicalization map.** The full canonicalization procedure (which includes expanding graph implementations) only runs *the first time the problem is canonicalized*. When the same problem is canonicalized in the future (*e.g.*, with new parameter values), the problem data $(A, b, c)$ can be obtained by multiplying a sparse matrix representing $C$ by the parameter vector (and reshaping); the adjoint of the derivative can be computed by just transposing the matrix. The naïve alternative — expanding graph implementations and extracting new problem data every time parameters are updated (and differentiating through this algorithm in the backward pass) — is much slower (see §7). The following lemma tells us that $C$ can be represented as a sparse matrix.

**Lemma 1.** *The canonicalizer map $C$ for a disciplined parametrized program can be represented with a sparse matrix $Q \in \mathbf{R}^{n \times p+1}$ and sparse tensor $R \in \mathbf{R}^{m \times n+1 \times p+1}$, where $m$ is the dimension of the constraints. Letting $\tilde{\theta} \in \mathbf{R}^{p+1}$ denote the concatenation of $\theta$ and the scalar offset $1$, the problem data can be obtained as $c = Q\tilde{\theta}$ and $[A \quad b] = \sum_{i=1}^{p+1} R_{[:,:,i]}\tilde{\theta}_i$.*

The proof is given in Appendix A.

### 4.3 Derivative of a conic solver

By applying the implicit function theorem [36, 34] to the optimality conditions of a cone program, it is possible to compute its derivative $\mathsf{D}s(A, b, c)$. To compute $\mathsf{D}^T s(A, b, c)$, we follow the methods presented in [1] and [4, §7.3]. Our calculations are given in Appendix B.

If the cone program is not differentiable at a solution, we compute a heuristic quantity, as is common practice in automatic differentiation [46, §14]. In particular, at non-differentiable points, a linear system that arises in the computation of the derivative might fail to be invertible. When this happens, we compute a least-squares solution to the system instead. See Appendix B for details.

### 4.4 Solution retrieval

The cone program obtained by canonicalizing a DPP-compliant problem uses the variable $\tilde{x} = (x, s) \in \mathbf{R}^n \times \mathbf{R}^k$, where $s \in \mathbf{R}^k$ is a slack variable. If $\tilde{x}^\star = (x^\star, s^\star)$ is optimal for the cone program, then $x^\star$ is optimal for the original problem (up to reshaping and scaling by a constant). As such, a solution to the original problem can be obtained by slicing, *i.e.*, $R(\tilde{x}^\star) = x^\star$. This map is evidently linear.

## 5 Implementation

We have implemented DPP and the reduction to ASA form in version 1.1 of CVXPY, a Python-embedded DSL for convex optimization [29, 3]; our implementation extends CVXCanon, an open-source library that reduces affine expression trees to matrices [62]. We have also implemented differentiable convex optimization layers in PyTorch and TensorFlow 2.0. These layers implement the forward and backward maps described in §4; they also efficiently support batched inputs (see §7).

We use the the `diffcp` package [1] to obtain derivatives of cone programs. We modified this package for performance: we ported much of it from Python to C++, added an option to compute the derivative using a dense direct solve, and made the forward and backward passes amenable to parallelization.

Our implementation of DPP and ASA form, coupled with our PyTorch and TensorFlow layers, makes our software the first DSL for differentiable convex optimization layers. Our software is open-source. CVXPY and our layers are available at

<center>https://www.cvxpy.org, https://www.github.com/cvxgrp/cvxpylayers.</center>

**Example.** Below is an example of how to specify the problem (4) using CVXPY 1.1.

```
1  import cvxpy as cp
2
3  m, n = 20, 10
4  x = cp.Variable((n, 1))
5  F = cp.Parameter((m, n))
6  g = cp.Parameter((m, 1))
7  lambd = cp.Parameter((1, 1), nonneg=True)
8  objective_fn = cp.norm(F @ x - g) + lambd * cp.norm(x)
9  constraints = [x >= 0]
10 problem = cp.Problem(cp.Minimize(objective_fn), constraints)
11 assert problem.is_dpp()
```

The below code shows how to use our PyTorch layer to solve and backpropagate through `problem` (the code for our TensorFlow layer is almost identical; see Appendix D).

<center>6</center>

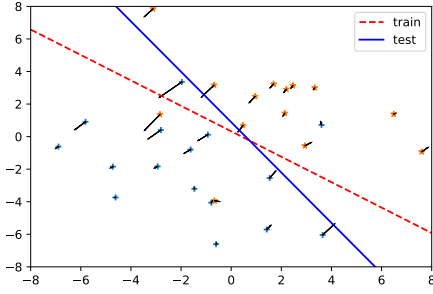
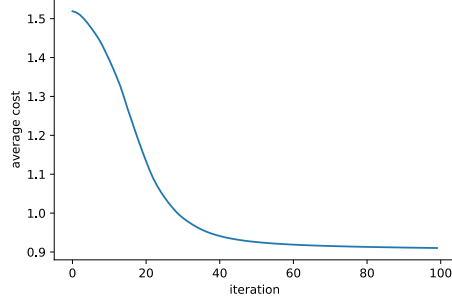

Figure 1: Gradients (black lines) of the logistic test loss with respect to the training data.

Figure 2: Per-iteration cost while learning an ADP policy for stochastic control.

```
1  import torch
2  from cvxpylayers.torch import CvxpyLayer
3
4  F_t = torch.randn(m, n, requires_grad=True)
5  g_t = torch.randn(m, 1, requires_grad=True)
6  lambd_t = torch.rand(1, 1, requires_grad=True)
7  layer = CvxpyLayer(
8      problem, parameters=[F, g, lambd], variables=[x])
9  x_star, = layer(F_t, g_t, lambd_t)
10 x_star.sum().backward()
```

Constructing `layer` in line 7-8 canonicalizes `problem` to extract $C$ and $R$, as described in §4.2. Calling `layer` in line 9 applies the map $R \circ s \circ C$ from §4, returning a solution to the problem. Line 10 computes the gradients of summing `x_star`, with respect to `F_t`, `g_t`, and `lambd_t`.

# 6 Examples

In this section, we present two applications of differentiable convex optimization, meant to be suggestive of possible use cases for our layer. We give more examples in Appendix E.

## 6.1 Data poisoning attack

We are given training data $(x_i, y_i)_{i=1}^N$, where $x_i \in \mathbf{R}^n$ are feature vectors and $y_i \in \{0, 1\}$ are the labels. Suppose we fit a model for this classification problem by solving

$$\text{minimize} \quad \tfrac{1}{N} \sum_{i=1}^N \ell(\theta; x_i, y_i) + r(\theta), \tag{6}$$

where the loss function $\ell(\theta; x_i, y_i)$ is convex in $\theta \in \mathbf{R}^n$ and $r(\theta)$ is a convex regularizer. We hope that the test loss $\mathcal{L}^{\text{test}}(\theta) = \tfrac{1}{M} \sum_{i=1}^M \ell(\theta; \tilde{x}_i, \tilde{y}_i)$ is small, where $(\tilde{x}_i, \tilde{y}_i)_{i=1}^M$ is our test set.

Assume that our training data is subject to a data poisoning attack [18, 49], before it is supplied to us. The adversary has full knowledge of our modeling choice, meaning that they know the form of (6), and seeks to perturb the data to maximally increase our loss on the test set, to which they also have access. The adversary is permitted to apply an additive perturbation $\delta_i \in \mathbf{R}^n$ to each of the training points $x_i$, with the perturbations satisfying $\|\delta_i\|_\infty \leq 0.01$.

Let $\theta^\star$ be optimal for (6). The gradient of the test loss with respect to a training data point, $\nabla_{x_i} \mathcal{L}^{\text{test}}(\theta^\star))$.gives the direction in which the point should be moved to achieve the greatest increase in test loss. Hence, one reasonable adversarial policy is to set $x_i := x_i + (.01)\mathbf{sign}(\nabla_{x_i} \mathcal{L}^{\text{test}}(\theta^\star))$. The quantity $(0.01) \sum_{i=1}^N \|\nabla_{x_i} \mathcal{L}^{\text{test}}(\theta^\star)\|_1$ is the predicted increase in our test loss due to the poisoning.

**Numerical example.** We consider 30 training points and 30 test points in $\mathbf{R}^2$, and we fit a logistic model with elastic-net regularization. This problem can be written using DPP, with $x_i$ as parameters

Table 1: Time (ms) to canonicalize examples, across 10 runs.

|  | Logistic regression | Stochastic control |
|---|---|---|
| CVXPY 1.0.23 | $18.9 \pm 1.75$ | $12.5 \pm 0.72$ |
| CVXPY 1.1 | $1.49 \pm 0.02$ | $1.39 \pm 0.02$ |

(see Appendix C for the code). We used our convex optimization layer to fit this model and obtain the gradient of the test loss with respect to the training data. Figure 1 visualizes the results. The orange ($\star$) and blue (+) points are training data, belonging to different classes. The red line (dashed) is the hyperplane learned by fitting the the model, while the blue line (solid) is the hyperplane that minimizes the test loss. The gradients are visualized as black lines, attached to the data points. Moving the points in the gradient directions torques the learned hyperplane away from the optimal hyperplane for the test set.

## 6.2 Convex approximate dynamic programming

We consider a stochastic control problem of the form

$$
\begin{aligned}
\text{minimize} \quad & \lim_{T \to \infty} \mathbb{E} \left[ \tfrac{1}{T} \sum_{t=0}^{T-1} \|x_t\|_2^2 + \|\phi(x_t)\|_2^2 \right] \\
\text{subject to} \quad & x_{t+1} = Ax_t + B\phi(x_t) + \omega_t, \quad t = 0, 1, \ldots,
\end{aligned}
\tag{7}
$$

where $x_t \in \mathbf{R}^n$ is the state, $\phi : \mathbf{R}^n \to \mathcal{U} \subseteq \mathbf{R}^m$ is the policy, $\mathcal{U}$ is a convex set representing the allowed set of controls, and $\omega_t \in \Omega$ is a (random, i.i.d.) disturbance. Here the variable is the policy $\phi$, and the expectation is taken over disturbances and the initial state $x_0$. If $\mathcal{U}$ is not an affine set, then this problem is in general very difficult to solve [50, 13].

**ADP policy.** A common heuristic for solving (7) is approximate dynamic programming (ADP), which parametrizes $\phi$ and replaces the minimization over functions $\phi$ with a minimization over parameters. In this example, we take $\mathcal{U}$ to be the unit ball and we represent $\phi$ as a quadratic *control-Lyapunov* policy [74]. Evaluating $\phi$ corresponds to solving the SOCP

$$
\begin{aligned}
\text{minimize} \quad & u^T P u + x_t^T Q u + q^T u \\
\text{subject to} \quad & \|u\|_2 \le 1,
\end{aligned}
\tag{8}
$$

with variable $u$ and parameters $P$, $Q$, $q$, and $x_t$. We can run stochastic gradient descent (SGD) on $P$, $Q$, and $q$ to approximately solve (7), which requires differentiating through (8). Note that if $u$ were unconstrained, (7) could be solved exactly, via linear quadratic regulator (LQR) theory [50]. The policy (8) can be written using DPP (see Appendix C for the code).

**Numerical example.** Figure 2 plots the estimated average cost for each iteration of gradient descent for a numerical example, with $x \in \mathbf{R}^2$ and $u \in \mathbf{R}^3$, a time horizon of $T = 25$, and a batch size of 8. We initialize our policy's parameters with the LQR solution, ignoring the constraint on $u$. This method decreased the average cost by roughly 40%.

## 7 Evaluation

Our implementation substantially lowers the barrier to using convex optimization layers. Here, we show that our implementation substantially reduces canonicalization time. Additionally, for dense problems, our implementation is competitive (in execution time) with a specialized solver for QPs; for sparse problems, our implementation is much faster.

**Canonicalization.** Table 1 reports the time it takes to canonicalize the logistic regression and stochastic control problems from §6, comparing CVXPY version 1.0.23 with CVXPY 1.1. Each canonicalization was performed on a single core of an unloaded Intel i7-8700K processor. We report the average time and standard deviation across 10 runs, excluding a warm-up run. Our extension achieves on average an order-of-magnitude speed-up since computing $C$ via a sparse matrix multiply is much more efficient than going through the DSL.

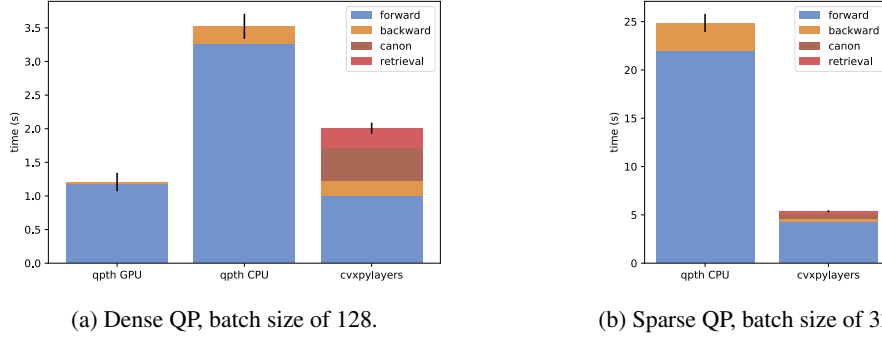

(a) Dense QP, batch size of 128.　　　　　　　(b) Sparse QP, batch size of 32.

Figure 3: Comparison of our PyTorch `CvxpyLayer` to `qpth`, over 10 trials. For `cvxpylayers`, we separate out the canonicalization and solution retrieval times, to allow for a fair comparison.

**Comparison to specialized layers.** We have implemented a batched solver and backward pass for our differentiable CVXPY layer that makes it competitive with the batched QP layer `qpth` from [6]. Figure 3 compares the runtimes of our PyTorch `CvxpyLayer` and `qpth` on a dense and sparse QP. The sparse problem is too large for `qpth` to run in GPU mode. The QPs have the form

$$
\begin{aligned}
\text{minimize} \quad & \tfrac{1}{2}x^T Q x + p^T x \\
\text{subject to} \quad & Ax = b, \\
& Gx \leq h,
\end{aligned}
\tag{9}
$$

with variable $x \in \mathbf{R}^n$, and problem data $Q \in \mathbf{R}^{n \times n}$, $p \in \mathbf{R}^n$, $A \in \mathbf{R}^{m \times n}$, $b \in \mathbf{R}^m$, $G \in \mathbf{R}^{p \times n}$, and $h \in \mathbf{R}^p$. The dense QP has $n = 128$, $m = 0$, and $p = 128$. The sparse QP has $n = 1024$, $m = 1024$, and $p = 1024$ and $Q$, $A$, and $G$ each have 1% nonzeros (See Appendix E for the code). We ran this experiment on a machine with a 6-core Intel i7-8700K CPU, 32 GB of memory, and an Nvidia GeForce 1080 TI GPU with 11 GB of memory.

Our implementation is competitive with `qpth` for the dense QP, even on the GPU, and roughly 5 times faster for the sparse QP. Our backward pass for the dense QP uses our extension to `diffcp`; we explicitly materialize the derivatives of the cone projections and use a direct solve. Our backward pass for the sparse QP uses sparse operations and LSQR [65], significantly outperforming `qpth` (which cannot exploit sparsity). Our layer runs on the CPU, and implements batching via Python multi-threading, with a parallel for loop over the examples in the batch for both the forward and backward passes. We used 12 threads for our experiments.

## 8 Discussion

**Other solvers.** Solvers that are specialized to subclasses of convex programs are often faster than more general conic solvers. For example, one might use OSQP [69] to solve QPs, or gradient-based methods like L-BFGS [54] or SAGA [28] for empirical risk minimization. Because CVXPY lets developers add specialized solvers as additional back-ends, our implementation of DPP and ASA form can be easily extended to other problem classes. We plan to interface QP solvers in future work.

**Nonconvex problems.** It is possible to differentiate through nonconvex problems, either analytically [37, 67, 5] or by unrolling SGD [33, 14, 61, 41, 70, 23, 38], Because convex programs can typically be solved efficiently and to high accuracy, it is preferable to use convex optimization layers over nonconvex optimization layers when possible. This is especially true in the setting of low-latency inference. The use of differentiable nonconvex programs in end-to-end learning pipelines, discussed in [42], is an interesting direction for future research.

## Acknowledgments

We gratefully acknowledge discussions with Eric Chu, who designed and implemented a code generator for SOCPs [26, 25], Nicholas Moehle, who designed and implemented a basic version of a code generator for convex optimization in unpublished work, and Brendan O'Donoghue. We also would like to thank the anonymous reviewers, who provided us with useful suggestions that improved the paper. S. Barratt is supported by the National Science Foundation Graduate Research Fellowship under Grant No. DGE-1656518.

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
