[Supplementary Material]

# A The canonicalization map

In this appendix, we provide a proof of Lemma 1. We compute $Q$ and $R$ via a reduction on the affine expression trees that represent the canonicalized problem. Let $f$ be the root node with arguments (descendants) $g_1, \ldots, g_n$. Then we obtain tensors $T_1, \ldots, T_n$ representing the (linear) action of $f$ on each argument. We recurse on each subtree $g_i$ and obtain tensors $S_1, \ldots, S_n$. Due to the DPP rules, for $i = 1, \ldots, n$, we either have $(T_i)_{j,k,\ell} = 0$ for $\ell \neq p + 1$ or $(S_i)_{j,k,\ell} = 0$ for $\ell \neq p + 1$. We define an operation $\psi(T_i, S_i)$ such that in the first case, $\psi(T_i, S_i) = \sum_{\ell=1}^{p+1}(T_i)_{[:,:,p+1]}(S_i)_{[:,:,\ell]}$, and in the second case $\psi(T_i, S_i) = \sum_{\ell=1}^{p+1}(T_i)_{[:,:,\ell]}(S_i)_{[:,:,p+1]}$. The tree rooted at $f$ then evaluates to $S_0 = \psi(T_1, S_1) + \cdots + \psi(T_n, S_n)$.

The base case of the recursion corresponds to the tensors produced when a variable, parameter, or constant node is evaluated. (These are the leaf nodes of an affine expression tree.)

- A variable leaf $x \in \mathbf{R}^d$ produces a tensor $T \in \mathbf{R}^{d \times n+1 \times 1}$, where $T_{i,j,1} = 1$ if $i$ maps to $j$ in the vector containing all variables, 0 otherwise.

- A parameter leaf $p \in \mathbf{R}^d$ produces a tensor $T \in \mathbf{R}^{d \times 1 \times p+1}$, where $T_{i,1,j} = 1$ if $i$ maps to $j$ in the vector containing all parameters, 0 otherwise.

- A constant leaf $c \in \mathbf{R}^d$ produces a tensor $T \in \mathbf{R}^{d \times 1 \times 1}$, where $T_{i,1,1} = c_i$ for $i = 1, \ldots, d$.

# B Derivative of a cone program

In this appendix, we show how to differentiate through a cone program. We first present some preliminaries.

**Primal-dual form of a cone program.** A (convex) cone program is given by

$$
\begin{array}{llll}
\text{(P)} & \text{minimize} & c^T x & \\
& \text{subject to} & Ax + s = b & \\
& & s \in \mathcal{K},
\end{array}
\qquad
\begin{array}{lll}
\text{(D)} & \text{minimize} & b^T y \\
& \text{subject to} & A^T y + c = 0 \\
& & y \in \mathcal{K}^*.
\end{array}
\qquad (10)
$$

Here $x \in \mathbf{R}^n$ is the *primal* variable, $y \in \mathbf{R}^m$ is the *dual* variable, and $s \in \mathbf{R}^m$ is the primal *slack* variable. The set $\mathcal{K} \subseteq \mathbf{R}^m$ is a nonempty, closed, convex cone with *dual cone* $\mathcal{K}^* \subseteq \mathbf{R}^m$. We call $(x, y, s)$ a solution of the primal-dual cone program (10) if it satisfies the KKT conditions:

$$
Ax + s = b, \quad A^T y + c = 0, \quad s \in \mathcal{K}, \quad y \in \mathcal{K}^*, \quad s^T y = 0.
$$

Every convex optimization problem can be reformulated as a convex cone program.

**Homogeneous self-dual embedding.** The homogeneous self-dual embedding reduces the process of solving (10) to finding a zero of a certain residual map [76]. Letting $N = n + m + 1$, the embedding uses the variable $z \in \mathbf{R}^N$, which we partition as $(u, v, w) \in \mathbf{R}^n \times \mathbf{R}^m \times \mathbf{R}$. The *normalized residual map* introduced in [24] is the function $\mathcal{N} : \mathbf{R}^N \times \mathbf{R}^{N \times N} \to \mathbf{R}^N$, defined by

$$
\mathcal{N}(z, Q) = \big((Q - I)\Pi + I\big)(z/|w|),
$$

where $\Pi$ denotes projection onto $\mathbf{R}^n \times \mathcal{K}^* \times \mathbf{R}_+$, and $Q$ is the skew-symmetric matrix

$$
Q = \begin{bmatrix} 0 & A^T & c \\ -A & 0 & b \\ -c^T & -b^T & 0 \end{bmatrix}. \qquad (11)
$$

If $\mathcal{N}(z, Q) = 0$ and $w > 0$, we can use $z$ to construct a solution of the primal-dual pair (10) as

$$
(x, y, s) = (u, \Pi_{\mathcal{K}^*}(v), \Pi_{\mathcal{K}^*}(v) - v)/w, \qquad (12)
$$

where $\Pi_{\mathcal{K}^*}(v)$ denotes the projection of $v$ onto $\mathcal{K}^*$. From here onward, we assume that $w = 1$. (If this is not the case, we can scale $z$ such that it is the case.)

**Differentiation.** A conic solver is a numerical algorithm for solving (10). We can view a conic solver as a function $\psi : \mathbf{R}^{m \times n} \times \mathbf{R}^m \times \mathbf{R}^n \to \mathbf{R}^{n+2m}$ mapping the problem data $(A, b, c)$ to a solution $(x, y, s)$. (We assume that the cone $\mathcal{K}$ is fixed.) In this section we derive expressions for the derivative of $\psi$, assuming that $\mathcal{S}$ is in fact differentiable. Interlaced with our derivations, we describe how to numerically evaluate the adjoint of the derivative map, which is necessary for backpropagation.

Following [1] and [4, Section 7], we can express $\psi$ as the composition $\phi \circ s \circ Q$, where

- $Q : \mathbf{R}^{m \times n} \times \mathbf{R}^m \times \mathbf{R}^n \to \mathbf{R}^{N \times N}$ maps the problem data to $Q$, given by (11),
- $s : \mathbf{R}^{N \times N} \to \mathbf{R}^N$ solves the homogeneous self-dual embedding, which we can implicitly differentiate, and
- $\phi : \mathbf{R}^N \to \mathbf{R}^n \times \mathbf{R}^m \times \mathbf{R}^m$ maps $z$ to the primal-dual pair, given by (12).

To backpropagate through $\psi$, we need to compute the adjoint of the derivative of $\psi$ at $(A, b, c)$ applied to the vector $(\mathrm{d}x, \mathrm{d}y, \mathrm{d}s)$, or

$$(\mathrm{d}A, \mathrm{d}b, \mathrm{d}c) = \mathsf{D}^T \psi(A, b, c)(\mathrm{d}x, \mathrm{d}y, \mathrm{d}s) = \mathsf{D}^T Q(A, b, c) \mathsf{D}^T s(Q) \mathsf{D}^T \phi(z)(\mathrm{d}x, \mathrm{d}y, \mathrm{d}s).$$

Since our layer only outputs the primal solution $x$, we can simplify the calculation by taking $\mathrm{d}y = \mathrm{d}s = 0$. By (12),

$$\mathrm{d}z = \mathsf{D}^T \phi(z)(\mathrm{d}x, 0, 0) = \begin{bmatrix} \mathrm{d}x \\ 0 \\ -x^T \mathrm{d}x \end{bmatrix}.$$

We can compute $\mathsf{D}s(Q)$ by implicitly differentiating the normalized residual map:

$$\mathsf{D}s(Q) = -(\mathsf{D}_z \mathcal{N}(s(Q), Q))^{-1} \mathsf{D}_Q \mathcal{N}(s(Q), Q). \tag{13}$$

This gives

$$\mathrm{d}Q = \mathsf{D}^T s(Q) \mathrm{d}z = -(M^{-T} \mathrm{d}z) \Pi(z)^T,$$

where $M = (Q - I)\mathsf{D}\Pi(z) + I$. Computing $g = M^{-T}\mathrm{d}z$ via a direct method (*i.e.*, materializing $M$, factorizing it, and back-solving) can be impractical when $M$ is large. Instead, one might use a Krylov method like LSQR [65] to solve

$$\underset{g}{\text{minimize}} \quad \|M^T g - \mathrm{d}z\|_2^2, \tag{14}$$

which only requires multiplication by $M$ and $M^T$. Instead of computing $\mathrm{d}Q$ as an outer product, we only obtain its nonzero entries. Finally, partitioning $\mathrm{d}Q$ as

$$\mathrm{d}Q = \begin{bmatrix} \mathrm{d}Q_{11} & \mathrm{d}Q_{12} & \mathrm{d}Q_{13} \\ \mathrm{d}Q_{21} & \mathrm{d}Q_{22} & \mathrm{d}Q_{23} \\ \mathrm{d}Q_{31} & \mathrm{d}Q_{32} & \mathrm{d}Q_{33} \end{bmatrix},$$

we obtain

$$\begin{aligned} \mathrm{d}A &= -\mathrm{d}Q_{12}^T + \mathrm{d}Q_{21} \\ \mathrm{d}b &= -\mathrm{d}Q_{23} + \mathrm{d}Q_{32}^T \\ \mathrm{d}c &= -\mathrm{d}Q_{13} + \mathrm{d}Q_{31}^T. \end{aligned}$$

**Non-differentiability.** To implicitly differentiate the solution map in (13), we assumed that the $M$ was invertible. When $M$ is not invertible, we approximate $\mathrm{d}Q$ as $-g^{\mathrm{ls}}\Pi(z)^T$, where $g^{\mathrm{ls}}$ is a least-squares solution to (14).

## C  Examples

This appendix includes code for the examples presented in §6.

**Logistic regression.**    The code for the logistic regression problem is below:

```
1  import cvxpy as cp
2  from cvxpylayers.torch import CvxpyLayer
3
4  beta = cp.Variable((n, 1))
5  b = cp.Variable((1, 1))
6  X = cp.Parameter((N, n))
7
8  log_likelihood = (1. / N) * cp.sum(
9      cp.multiply(Y, X @ beta + b) - cp.logistic(X @ beta + b)
10 )
11 regularization = -0.1 * cp.norm(beta, 1) -0.1 *
      cp.sum_squares(beta)
12
13 prob = cp.Problem(cp.Maximize(log_likelihood + regularization))
14 fit_logreg = CvxpyLayer(prob, parameters=[X], variables=[beta,
      b])
```

**Stochastic control.**    The code for the stochastic control problem (7) is below:

```
1  import cvxpy as cp
2  from cvxpylayers.torch import CvxpyLayer
3
4  x_cvxpy = cp.Parameter((n, 1))
5  P_sqrt_cvxpy = cp.Parameter((m, m))
6  P_21_cvxpy = cp.Parameter((n, m))
7  q_cvxpy = cp.Parameter((m, 1))
8
9  u_cvxpy = cp.Variable((m, 1))
10 y_cvxpy = cp.Variable((n, 1))
11
12 objective = .5 * cp.sum_squares(P_sqrt_cvxpy @ u_cvxpy) +
      x_cvxpy.T @ y_cvxpy + q_cvxpy.T @ u_cvxpy
13 prob = cp.Problem(cp.Minimize(objective),
14   [cp.norm(u_cvxpy) <= .5, y_cvxpy == P_21_cvxpy @ u_cvxpy])
15
16 policy = CvxpyLayer(prob,
17   parameters=[x_cvxpy, P_sqrt_cvxpy, P_21_cvxpy, q_cvxpy],
18   variables=[u_cvxpy])
```

## D  TensorFlow layer

In §5, we showed how to implement the problem (4) using our PyTorch layer. The below code shows how to implement the same problem using our TensorFlow 2.0 layer.

```
1  import tensorflow as tf
2  from cvxpylayers.tensorflow import CvxpyLayer
3
4  F_t = tf.Variable(tf.random.normal(F.shape))
5  g_t = tf.Variable(tf.random.normal(g.shape))
6  lambd_t = tf.Variable(tf.random.normal(lambd.shape))
7  layer = CvxpyLayer(problem, parameters=[F, g, lambd],
      variables=[x])
8  with tf.GradientTape() as tape:
9    x_star, = layer(F_t, g_t, lambd_t)
10 dF, dg, dlambd = tape.gradient(x_star, [F_t, g_t, lambd_t])
```

# E   Additional examples

In this appendix we provide additional examples of constructing differentiable convex optimization layers using our implementation. We present the implementation of common neural networks layers, even though analytic solutions exist for some of these operations. These layers can be modified in simple ways such that they do *not* have analytical solutions. In the below problems, the optimization variable is $y$ (unless stated otherwise). We also show how prior work on differentiable convex optimization layers such as OptNet [6] is captured by our framework.

The **ReLU**, defined by $f(x) = \max\{0, x\}$, can be interpreted as projecting a point $x \in \mathbf{R}^n$ onto the non-negative orthant as

$$\begin{array}{ll} \text{minimize} & \frac{1}{2}\|x - y\|_2^2 \\ \text{subject to} & y \geq 0. \end{array}$$

We can implement this layer with:

```
x = cp.Parameter(n)
y = cp.Variable(n)
obj = cp.Minimize(cp.sum_squares(x-y))
cons = [y >= 0]
prob = cp.Problem(obj, cons)
layer = CvxpyLayer(prob, parameters=[x], variables=[y])
```

The **sigmoid** or **logistic** function, defined by $f(x) = (1 + e^{-x})^{-1}$, can be interpreted as projecting a point $x \in \mathbf{R}^n$ onto the interior of the unit hypercube as

$$\begin{array}{ll} \text{minimize} & -x^\top y - H_b(y) \\ \text{subject to} & 0 < y < 1, \end{array}$$

where $H_b(y) = -\left(\sum_i y_i \log y_i + (1 - y_i) \log(1 - y_i)\right)$ is the binary entropy function. This is proved, *e.g.*, in [4, Section 2.4]. We can implement this layer with:

```
x = cp.Parameter(n)
y = cp.Variable(n)
obj = cp.Minimize(-x.T*y - cp.sum(cp.entr(y) + cp.entr(1.-y)))
prob = cp.Problem(obj)
layer = CvxpyLayer(prob, parameters=[x], variables=[y])
```

The **softmax**, defined by $f(x)_j = e^{x_j} / \sum_i e^{x_i}$, can be interpreted as projecting a point $x \in \mathbf{R}^n$ onto the interior of the $(n-1)$-simplex $\Delta_{n-1} = \{p \in \mathbf{R}^n \mid 1^\top p = 1 \text{ and } p \geq 0\}$ as

$$\begin{array}{ll} \text{minimize} & -x^\top y - H(y) \\ \text{subject to} & 0 < y < 1, \\ & 1^\top y = 1, \end{array}$$

where $H(y) = -\sum_i y_i \log y_i$ is the entropy function. This is proved, *e.g.*, in [4, Section 2.4]. We can implement this layer with:

```
x = cp.Parameter(d)
y = cp.Variable(d)
obj = cp.Minimize(-x.T*y - cp.sum(cp.entr(y)))
cons = [sum(y) == 1.]
prob = cp.Problem(obj, cons)
layer = CvxpyLayer(prob, parameters=[x], variables=[y])
```

The **sparsemax** [58] does a Euclidean projection onto the simplex as

$$\begin{array}{ll} \text{minimize} & ||x - y||_2^2 \\ \text{subject to} & 1^\top y = 1, \\ & 0 \le y \le 1. \end{array}$$

We can implement this layer with:

```
1 x = cp.Parameter(n)
2 y = cp.Variable(n)
3 obj = cp.sum_squares(x-y)
4 cons = [cp.sum(y) == 1, 0. <= y, y <= 1.]
5 prob = cp.Problem(cp.Minimize(obj), cons)
6 layer = CvxpyLayer(prob, [x], [y])
```

The **constrained softmax** [59] solves the optimization problem

$$\begin{array}{ll} \text{minimize} & -x^\top y - H(y) \\ \text{subject to} & 1^\top y = 1, \\ & y \le u, \\ & 0 < y < 1. \end{array}$$

We can implement this layer with:

```
1 x = cp.Parameter(n)
2 y = cp.Variable(n)
3 obj = -x*y-cp.sum(cp.entr(y))
4 cons = [cp.sum(y) == 1., y <= u]
5 prob = cp.Problem(cp.Minimize(obj), cons)
6 layer = CvxpyLayer(prob, [x], [y])
```

The **constrained sparsemax** [55] solves the optimization problem

$$\begin{array}{ll} \text{minimize} & ||x - y||_2^2, \\ \text{subject to} & 1^\top y = 1, \\ & 0 \le y \le u. \end{array}$$

We can implement this layer with:

```
1 x = cp.Parameter(n)
2 y = cp.Variable(n)
3 obj = cp.sum_squares(x-y)
4 cons = [cp.sum(y) == 1., 0. <= y, y <= u]
5 prob = cp.Problem(cp.Minimize(obj), cons)
6 layer = CvxpyLayer(prob, [x], [y])
```

The **Limited Multi-Label (LML)** layer [7] solves the optimization problem

$$\begin{array}{ll} \text{minimize} & -x^\top y - H_b(y) \\ \text{subject to} & 1^\top y = k, \\ & 0 < y < 1. \end{array}$$

We can implement this layer with:

```
1 x = cp.Parameter(n)
2 y = cp.Variable(n)
3 obj = -x*y-cp.sum(cp.entr(y))-cp.sum(cp.entr(1.-y))
4 cons = [cp.sum(y) == k]
5 prob = cp.Problem(cp.Minimize(obj), cons)
6 layer = CvxpyLayer(prob, [x], [y])
```

**The OptNet QP.**   We can re-implement the OptNet QP layer [6] in a few lines of code. The OptNet layer is a solution to a convex quadratic program of the form

$$\begin{aligned}
\text{minimize} \quad & \tfrac{1}{2}x^\top Q x + q^\top x \\
\text{subject to} \quad & Ax = b, \\
& Gx \le h,
\end{aligned}$$

where $x \in \mathbf{R}^n$ is the optimization variable, and the problem data are $Q \in \mathbf{R}^{n \times n}$ (which is positive semidefinite), $p \in \mathbf{R}^n$, $A \in \mathbf{R}^{m \times n}$, $b \in \mathbf{R}^m$, $G \in \mathbf{R}^{p \times n}$, and $h \in \mathbf{R}^p$. We can implement this with:

```
1  Q_sqrt = cp.Parameter((n, n))
2  q = cp.Parameter(n)
3  A = cp.Parameter((m, n))
4  b = cp.Parameter(m)
5  G = cp.Parameter((p, n))
6  h = cp.Parameter(p)
7  x = cp.Variable(n)
8  obj = cp.Minimize(0.5*cp.sum_squares(Q_sqrt*x) + q.T @ x)
9  cons = [A @ x == b, G @ x <= h]
10 prob = cp.Problem(obj, cons)
11 layer = CvxpyLayer(prob, parameters=[Q_sqrt, q, A, b, G, h],
       variables=[x])
```

Note that we take the matrix square-root of $Q$ in PyTorch, outside the CVXPY layer, to get the derivative with respect to $Q$. DPP does not allow the quadratic form atom to be parametrized, as discussed in §4.1.