[Reviews · NeurIPS 2019]

Reviewer 1



*** update after author feedback *** The author feedback proposes a clearer exposition of the ASA form and being more clear about what happens when the map is nondifferentiable, which would be welcome additions to the submission. With these modifications, and further reflection about the possible significance of this being a user-friendly tool, I have upgraded my assessment of the paper. *** end of update *** Originality: The paper is of modest originality. It augments the existing idea of differentiating through conic programs by implicitly differentiating the optimality conditions, so that the user can specify a problem in DCP form and still take advantage of this technology. Clarity: The paper is, for the most part, very clearly written, including nice devices like running examples to illustrate what is going on. Still a major issue for me was that when some of the key ideas were introduced in section 4, they were not explained clearly and systematically. The example I found most frustrating was the term "jointly DCP", in the sense of "jointly DCP in x (the variable) and theta (the parameter)". This appears as an essential component of the definition of ASA form, but is not defined itself. It is clearly not the case that this requires joint convexity in x and theta (from the examples) so what exactly does this mean? This really must be explained more clearly. Similarly, I found statements like "which ensures that the maps C and R are affine; in practice, this means that their derivatives are easy to compute" quite confusing. Does this "ensure that the maps C and R are affine" (which would then imply their derivative are easy to compute)? Or does it mean that C and R could be taken to be non-affine functions, as long as their derivatives are easy to compute? Or does it mean something else entirely? Quality: The paper appears, overall, to be technically sound. I found that the work was a little too dismissive of the situation in which the solution mapping is not differentiable. I think dealing with nondifferentiability is far beyond the scope of this work, but I think the authors would be better served to admit that there are many examples of convex optimization problems (e.g., linear programs) where the solution mappings are nondifferentiable functions of the problem data. Having said that, they are differentiable almost everywhere, so whether this is an issue or not depends on how the points at which the derivatives are evaluated are chosen. The citation [4, section 14] is a bit too broad, and doesn't sufficiently explain how the work under review deals with the nondifferentiable case. Does it issue a warning? Does it fail silently? A much more minor comment is that in a couple of places in the paper the authors asser that the solution of convex optimization problems can be obtained "exactly". (e.g., implicitly in line 82, "we compute it exactly") and in line 382. In these cases it would be reasonable to say that these things can be computed "to high accuracy" or something a little more vague, since "exact computation" is a much stronger statement, and for many convex optimization problems we do not know how to exactly compute the solution, and so would not know how to exactly compute the gradient of the solution mapping. Significance: From a technical point of view the contribution is not that significant---the ideas behind it are relatively simple, or are extensions of non-trivial ideas that have become "normalized" as they have matured. Having said that, I think this paper is of quite high significance in the sense that if it the implementation is done well and is easy to use then (possibly very many) researchers are likely to make direct use of this work.

Reviewer 2



--- after author response --- I thank the authors for the clarifications, and for the promise to improve the organization. --- original review --- The paper paints a clear high-level picture of differentiating through (disciplined) convex programs. DCP is a very useful tool in prototyping but also in practical applications, so it is in my opinion a good building block for generic differentiable convex optimization layers. Originality: 5/5 The paper relies on already-studied building blocks: the DCP formalism, and the residual approach to differentiation through a cone solver. However, combining these approaches to build a general-purpose differentiable convex optimization method is original and a valuable innovation. The ASA formalism, an extension of DCP which represents parameters via affine mappings, is novel and helpful for efficiency reasons. - Quality: 5/5 The paper is well supported. The two application examples illustrate very well the wide range of powerful things enabled by differentiable convex optimization. The code samples are a nice touch, showcasing how simple such tasks become using the proposed framework. Minor mistakes/typos: - at line 316, some (all?) of the cited work unroll *gradient descent*, not **stochastic** gradient - The variational form of softmax in supplementary material should also have a non-negativity constraint (it is there in the equations but not in the code) - Clarity: 4.5/5 At a high level the paper is very well written and smooth to read, very clear about its contributions. Some of the deeper, more technical constructions should be explained a bit clearer: - the ASA form (sec 4.1) should be clearly given constructively, rather than the vague phrasing "our rule has one important exception..." - related to above, I would suggest expanding the rigorous definition of DCP and the theorem at line 100 to get more emphasis. - The sparse canonicalization map: the example at lines 182-189 are pretty confusing. What are S, T, \psi and how to they relate with the desired Q & R? As it is, the example confused me more than it clarified my understanding. - Figure 1 lacks some explanation, what is the color of the markers? Consider using x/o instead of colors. Significance: 5/5 I believe overall this work is very significant. It is rather technical & engineering-based, but i believe it can have a high impact. DCP libraries like CVXPY are very useful in prototyping. A lot of recent work involves differentiating through optimization problems, and while for specific applications it is likely that specialized algorithms perform better, the proposed general approach will be very useful at least to kickstart research projects. The neural architectures one can build with convex optimization hidden layers are powerful and promising. Question: At line 209 you claim batching is done using a parallel loop. Are there settings where faster batching might be possible?

Reviewer 3



If I understand correctly, the layer is somehow like an auto-differentiation variant, that represents complex but disciplined convex problems as a tree of the composition of subprograms of bilinear, then apply a cone optimizer to generate the solution and differentiating over parameters to get the whole solution path, which is similar to LARS. I think the implementation is friendly enough for users like me, and I appreciate the efforts to make it a library. The authors did not mention the concept of the condition number of the convex problems, therefore I am curious whether the inner-most cone program is better-conditioned than the original program, theoretically or empirically. Besides, I am curious whether this applies to non-smooth subprograms as well, e.g. with $\ell_1$ norm or ReLU function insides, where subgradients have to be applied.

[Author Response · NeurIPS 2019]

We thank the reviewers for their constructive feedback on our paper. We especially appreciate our reviewers' conviction that our implementation will be a widely used tool for embedding convex optimization problems in end-to-end learning models. By combining DCP, the residual approach to differentiation through a cone solver, and the proposed ASA form for parameters, we have demonstrated a general approach for efficiently differentiating through convex optimization problems and incorporating these problems as layers within standard deep learning packages. We also share the reviewers' opinions that several aspects of the papers can be improved. We address some of the main reviewers concerns and comments below, and we will edit the final version of the paper to clarify all these points.

**Shared concerns.** Reviewers 1 and 2 found some of our explanations of ASA form and DPP difficult to follow. We agree, based upon the reviewer comments, that this section can be clarified, and we will make the following changes.

We will expand on the definition of DCP in §3 and will revise §4 to make the definitions of ASA form and DPP more constructive. We will also explain the motivation for our ruleset (reviewer 1's guess is essentially correct). We will provide additional examples of DPP-compliant and non-compliant expressions and problems. Space permitting (or in an appendix if needed), we will also include simple figures illustrating why one expression is DPP, and why another is not.

To give a brief idea of our proposed edits, the basic idea of DPP (which we will expand upon in our revision) is this: (1) first, every parameter appearing in an expression tree as `param * expr`, where `expr` does *not* contain parameters, is cast to a constant for the purpose of DCP analysis; then (2) to be DPP, the problem must be DCP if the remaining parameters were replaced with variables. This is what we meant by our vague phrasing "jointly DCP ... [with] one important exception". As a quick example, let `lambda` be a positive parameter and `x` a variable: the expression `lambda * norm(x) + exp(lambda)` is DPP because (1) in `lambda * norm(x)`, `lambda` is cast to a constant `const_lambda` because it multiplies a parameter-free expression and (2) `const_lambda * norm(x) + exp(lambda)` is DCP in both `x` and `lambda`.

We will separately explain how to reduce certain expressions in which parameters are multiplied together (*e.g.*, `param1 * param2 * param3 * variable`) to DPP-compliant expressions. This will allow us to simplify the definition of DPP (in particular, it will allow us to delete the rules outlined in lines 139-141), without sacrificing the expressiveness of our grammar. The reduction is currently given in lines 158-162; this reduction will be described as an optional transformation that the user can perform manually (but can be easily automated), before canonicalizing a DPP program.

**Reviewer 1.** ASA form means that $C$ and $R$ must be affine. We will clarify this point. We will also change all mentions of "exact solution" to "solve to high accuracy".

We hope that our brief explanation of DPP in the previous section sheds some light on why the quadratic form in lines 147-151 is not DPP. The expression `quad_form(x, P)`, where $x$ is a variable and $P$ is a parameter, is not DCP in $x$ and $P$ (with $P$ treated as a variable), hence it is not DPP. On the other hand, `square(norm(P*x))` is DPP, because the parameter $P$ is first cast to a constant (since it multiplies $x$), and `square(norm(P*x))` is DCP (since $P$ is treated as a constant). In the revision, we will make sure to clearly explain this.

We agree that we didn't highlight sufficiently the possible non-differentiability of the solution map. Like other layers, optimization layers can lead to undefined behavior due to non-differentiability, but, unlike other layers, it is not always obvious whether the solution map will be differentiable at a point. We strongly recommend for users to formulate their optimization layers so that the solution is unique and the derivative is well-defined, however we understand that this will not always be the case. We will add a further discussion of non-differentiability (including more specific citations) and how the library behaves in such cases. We follow the convention of recent automatic differentiation libraries (*e.g.*, PyTorch and Tensorflow), and return the gradient when it exists, and otherwise return something "reasonable." In our case we consider "reasonable" to be the least squares solution to the non-invertible linear system for the derivative.

**Reviewer 2.** We thank the reviewer for their appreciation of the significance of our contribution, most of all in enabling researchers to explore the applications of differentiable convex optimization layers. We will greatly clarify the exposition of ASA form and define it constructively (see shared concerns), revise the sparse canonicalization map example and Figure 1, and follow all minor suggestions.

As an alternative to multithreading, we have implemented batched computations on the GPU by providing a PyTorch callback to the conic solver SCS. Space and time permitting, we will discuss the tradeoffs of both approaches. Regardless, our open-source implementation will support both.

**Reviewer 3.** Our contribution is a new approach to differentiating through (smooth or nonsmooth) disciplined convex programs, considering convex programs as mappings from parameters to solution values. Computing a solution path, as in LARS, is a related but orthogonal task. Analysis of the condition number of the convex problems generated by these reductions is an interesting but orthogonal direction, and is unfortunately out of the scope of this paper.

[Meta-Review · NeurIPS 2019]

Congratulations --- the reviewers all agreed that this work was a creative and interesting contribution to the machine learning literature! In your revision, please address any questions from the reviews and promises in your rebuttal.